# A cross-sectional survey of burnout in a sample of resident physicians in Sudan

**Yasir Ahmed Mohammed Elhadi**[1,2]*, **Abdelmuniem Ahmed**[3], **Elhadi B. Salih**[4], **Osman S. Abdelhamed**[5], **Mohamed Hayder Hamid Ahmed**[6], **Noha Ahmed El Dabbah**[1]

**1** Department of Health Administration and Behavioral Sciences, High Institute of Public Health Alexandria University, Alexandria, Egypt, **2** Department of Public Health, Medical Research Office, Sudanese Medical Research Association, Khartoum, Sudan, **3** Physiology Department, Faculty of Medicine, University of Gezira, Wad Madani, Sudan, **4** Federal Ministry of Health, Khartoum, Sudan, **5** Faculty of Medicine, Ahfad University for Women, Omdurman, Sudan, **6** Sudanese Medical Specialization Board, Wad-Medani, Sudan

* hiph.yelhadi@alexu.edu.eg

## Abstract

### Background

Resident physicians in Sudan face a variety of physical and psychological stressors. Nevertheless, the prevalence of burnout syndrome among this critical population remains unknown. The purpose of this study was to estimate the prevalence rate of burnout and its associated factors in a sample of resident physicians in Sudan.

### Methods

A cross-sectional design was used to assess the burnout syndrome among resident physicians at the teaching hospitals of Wad-Medani in Gezira state, east-central Sudan. Three hundred resident physicians at the dermatology, general surgery, pediatrics, obstetrics and gynecology, psychiatry, ear, nose and throat (ENT), oncology, urology, and internal medicine departments, were approached and invited to participate in the study. The Arabic version of the Maslach Burnout Inventory was distributed to respondents from July to October 2021.

### Results

From the 300 resident physicians, 208 (69.3%) responded. The average age of the study population was 29.99 ± 3.01 years, with more than half were females (56.7%), single (59.6%), and with more than three years of residency experience (50.5%). In total, 86.1% met the criteria for burnout in at least one dimension and 13.9% in all three dimensions. On the dimension of emotional exhaustion (EE), 70.7% reported high levels of burnout. While, 44.2% reported high levels of depersonalization (DP), and 73.1% experienced a sense of decreased professional accomplishment (PA). There were significant differences in burnout, EE, and DP levels among different specialties, with the pediatrics-specialty trainees reported higher levels. Burnout syndrome was associated with the working hours per single duty; participants who reported working for more than 24 hours had experienced higher levels of burnout, EE, and DP.

**Data Availability Statement:** The datasets used and/or analyzed during the current study are available from the following DOI link: https://doi.org/10.6084/m9.figshare.19184831.

**Funding:** The author(s) received no specific funding for this work.

**Competing interests:** The authors have declared that no competing interests exist.

## Conclusion

Large-scale studies are required to assess the determinants of burnout syndrome among resident physicians in Sudan. In addition, Stakeholders should urgently implement effective remedies to protect the mental health of resident physicians.

## Introduction

Burnout syndromes (BOS) is a psychological syndrome emerging as a prolonged response to chronic interpersonal stressors on the job [1]. The term "burnout" was coined in the 1970s by the American psychologist Herbert Freudenberger [2]. In the 11[th] revision of the International Classification of Diseases (ICD-11), BOS was conceptualized as "a syndrome resulting from chronic workplace stress that has not been successfully managed. BOS refers specifically to phenomena in the occupational context and should not be applied to describe experiences in other areas of life" [3]. Three sub-dimensions characterize BOS; high emotional exhaustion, an increase in the mental distance from the profession; high depersonalization, skepticism about the career; and a sense of decreased professional achievement [1].

Evidence suggests that healthcare workers suffer from high levels of BOS [4–6], with trainee physicians being at increased risk of BOS compared to other healthcare workers [7, 8]. Among trainee physicians, several occupational and individual factors have been identified. However, it is still unclear which factors are the most significant in promoting BOS development [9]. Work-related factors vary across different medical departments, indicating specific fundamental disparities in working environments correlated with BOS among these populations [9]. BOS also appears to be influenced by payment models, with physicians who only receive an incentive or performance-based pay having a considerably higher burnout rate than the salary-paid physicians [10, 11]. BOS is influenced by the organizational context, such as negative leadership behaviors and inadequate interprofessional collaboration, as well as opportunities for growth and social support for physicians [12]. BOS is also linked to poor working conditions, increased work demands, work-life imbalance, postgraduate training requirements that interfere with personal life, and lack of senior support [9]. Physician burnout is financially costly and has many other consequences, such as leaving the medical career [13]. In addition, depression, marital complications, medical errors, substance abuse, and suicidal behavior might occur [14]. Such consequences have detrimental impacts on healthcare organizations, physicians, and the quality of patient care provided [13].

The Sudan Medical Specialization Board (SMSB) is the professional training organization responsible for managing and delivering medical and health-specialized programs in Sudan [15]. Sudan has difficulty coordinating human resources for health policies and overall health planning. There has been an imbalance in the training and production of health professionals in certain professions due to a lack of coordination between the health authorities–represented by the Federal Ministry of Health–and the medical training sector–represented by the SMSB [16]. Furthermore, the mass migration of qualified health workers due to the economic crisis has left Sudan with a severe shortage in terms of qualified trainers. Moreover, the privatization of the health sector during the previous regime in Sudan has made the remaining qualified practitioners much less available for education and training in teaching hospitals, affecting the quality of medical training [17]. In a recent curriculum analysis of the residency training program in Sudan, there were no specific educational or training plans identified [18]. Furthermore, a study assessed the perception of social support, role autonomy, and teaching among resident physicians in Sudan, in which most resident physicians complained about lack of

social support and role autonomy, lack of time allocated for learning sessions, and lack of research teaching and opportunities [19].

Despite being established in 1995, the SMSB has just started paying less than 100 $ per month to resident physicians in 2021. At the same time, doctors accepted by training programs must pay annual training fees to the SMSB, posing a significant financial burden on resident physicians and forcing them to work extra hours in private hospitals or clinics to cover their daily living expenses. Therefore, this affects their training quality and their time for academic achievement, which poses an additional risk for stress and BOS [20]. Depending on the specialty, the training period can last four to five years. Due to the SMSB's random distribution of trainees, a resident physician would be required to work in other Sudanese state hospitals for a significant portion of their training period, away from their homes and families. This scenario suggests extreme psychological and mental health impacts among resident physicians in Sudan, which necessitates urgent investigation to propose effective remedies.

To our knowledge and based on our search, there were no previous studies concerning the prevalence of BOS among this critical population in Sudan. This study was conducted in order to fulfill this research gap. The purpose of this study was to estimate the prevalence rate of burnout and its related factors in a sample of Sudanese resident physicians.

## Material and methods

### Study design, setting, and population

A cross-sectional survey design was followed in this study, conducted at the teaching hospitals of Wad-Medani district, Gezira state, east-central Sudan. There are nine teaching hospitals in Wad-Medani covering nine medical specialties, serving more than 3 million Sudanese population, and affiliated with the University of Gezira-Faculty of Medicine, were selected to conduct the current study. At the time of data collection, the 300 resident physicians at the dermatology, general surgery, pediatrics, obstetrics and gynecology, psychiatry, ear nose and throat (ENT), oncology, urology, and internal medicine departments, were approached and invited to participate in the study.

### Data collection

Data were collected from July to October 2021 via a self-administered questionnaire comprised of two sections. The first section included the informed consent and other items related to participants' sociodemographic and work-related attributes, including age, sex, marital status, specialty, period of residency experience, and working hours/duty.

The second section included the Arabic validated version of the Maslach Burnout Inventory Human Services Survey (MBI-HSS) [21, 22], used after obtaining the required permissions from *Mind Garden*, Inc., an independent publisher of tools and instruments for psychological assessments of BOS, anxiety, and leadership among others [23]. The MBI-HSS is the most accepted and widely used instrument for assessing BOS [24]. It consists of 22 items investigating the three dimensions of BOS; nine questions for emotional exhaustion (EE); five questions for depersonalization (DP); and eight questions for personal achievement (PA). Each item was rated on a seven-point frequency rating scale, ranging from never (score 0) to every day (score 6). Higher scores on the EE and DP subscales were associated with higher levels of BOS, whereas a high level of PA was associated with lower levels of BOS.

### Search strategy

Similarly, in other cross-sectional studies of BOS among healthcare workers [25], we performed an online data search to compare the burnout rate between resident physicians in

Sudan and other countries. We searched the PubMed database for similar cross-specialty studies of BOS among resident physicians published between 2000–2021. Two authors (Y.E.) and (A.A.) performed the search. We excluded studies reporting the BOS rate among single-specialty resident physicians. The authors evaluated the search results, and relevant studies were included and extracted on an excel sheet. The reported burnout rate in each of the included studies was compared to the current study's results.

## Statistical analysis

Data were analyzed using IBM SPSS software package version 20.0. The Kolmogorov-Smirnov test was utilized to ensure the normal distribution of variables. The three subscales measuring burnout syndrome were categorized according to the scoring system of MBI-HSS [26]. The Cronbach's Alpha for the MBI-HSS in this study was 0.79, indicating the high reliability of the overall measurement. For normally distributed quantitative variables, the Student's t-test was used to compare two groups, while ANOVA was used to compare more than two groups. The significance of the results obtained was determined at the 5% level of alpha error.

## Ethical considerations

This study was performed in accordance with the ethical standards laid down in the 1964 Declaration of Helsinki and its later amendments or comparable ethical standards [27]. The study was approved by the Health Sector Ethical Review Committee, University of Gezira (IRB approval No: 00036–21). Data anonymity and confidentiality were guaranteed. All participants provided written consent to participate, which was included in the data collection tool. Before the start of the study, it was made clear that participants had the right to withdraw at any time.

## Results

### Sociodemographic and work-related characteristics of respondents

The sociodemographic and work-related characteristics of respondents are shown in Table 1. A total of 208 resident physicians with a mean age of 29.99 ± 3.01 years participated in the present study (response rate = 69.3%). More than half of respondents were females (56.7%), single (59.6%), and with more than three years of residency experience (50.5%). Most of the participants were medicine specialty residents (29.8%). More than one-third reported working for more than 24 hours per single duty (32.7%).

### The prevalence and levels of burnout syndrome among participants

The prevalence rate of BOS and its dimensions are displayed in Table 2. More than two-thirds (70.7%) of resident physicians who participated in the present study reported high levels of burnout on the dimension of emotional exhaustion, while 44.2% reported high levels of depersonalization, and 73.1% experienced a sense of decreased professional accomplishment. Overall, 86.1% met the criteria for BOS in at least one dimension and 13.9% in all three dimensions.

### Factors associated with burnout syndrome among participants

Parameters associated with the BOS are shown in Table 3. There are significant differences in the levels of emotional exhaustion (EE), depersonalization (DP), and the overall burnout among different specialties, with the pediatrics-specialty resident physicians, demonstrated higher overall burnout [mean ± SD, 3.19 ± 0.86 (P<0.001)], higher EE [mean ± SD, 4.36 ± 1.26(P<0.001)], and higher DP [mean ± SD, 3.09 ± 1.32) (P<0.001)]. The overall burnout, EE, and DP were significantly associated with the working hours per single duty;

**Table 1. Sociodemographic and work-related characteristics of respondents.**

| Sociodemographic characteristics | N | % |
|---|---|---|
| **Age (years)** | | |
| <30 | 104 | 50 |
| 30 –<35 | 84 | 40.4 |
| ≥35 | 20 | 9.6 |
| Mean ± Standard Deviation = 29.99 ± 3.01 | | |
| **Gender** | | |
| Male | 90 | 43.3 |
| Female | 118 | 56.7 |
| **Marital status** | | |
| Single | 124 | 59.6 |
| Married | 81 | 38.9 |
| Divorced/Widowed | 3 | 1.4 |
| **Specialty** | | |
| Medicine | 62 | 29.8 |
| Pediatrics | 38 | 18.3 |
| Urology | 13 | 6.3 |
| Obstetrics & Gynecology | 26 | 12.5 |
| Surgery | 38 | 18.3 |
| Ear Nose and Throat | 7 | 3.4 |
| Oncology | 9 | 4.3 |
| Dermatology | 13 | 6.3 |
| Psychiatry | 2 | 1 |
| **Years of residency experience** | | |
| <3 | 103 | 49.5 |
| ≥3 | 105 | 50.5 |
| Mean ± Standard Deviation = 2.46 ± 1.05 | | |
| **Working hours per duty** | | |
| 8 to11 hours | 37 | 17.8 |
| 12 to 17 hours | 50 | 24 |
| 18 to 24 hours | 53 | 25.5 |
| More than 24 hours | 68 | 32.7 |

**N:** Number.

physicians working for more than 24 hours per single duty have experienced higher levels of BOS, EE, and DP (P<0.001). However, no significant differences were detected in the levels of professional accomplishment related to respondents' sociodemographic and work-related characteristics. Additionally, there were no significant differences observed in the levels of burnout and its three dimensions among different age groups, gender, years of residency experience, or marital status of resident physicians (Table 3).

## Burnout rate among resident physicians in Sudan compared to other countries

The database search generated 720 potentially relevant articles. After the title, abstract, and full texts screening, only 11 studies were included. The sample size of resident physicians in the included studies ranged from 68 in the United States (USA) to 3350 in Syria. All included studies have reported a higher prevalence of burnout syndrome than the current study. However, resident physicians in Sudan experienced higher emotional exhaustion and depersonalization

**Table 2. The frequency of resident physicians by levels of burnout in the three dimensions.**

| Burnout dimensions | Low | Moderate | High |
|---|---|---|---|
| **Emotional exhaustion (EE)**[a] | 28 (13.5%) | 33 (15.9%) | 147 (70.7%) |
| **Depersonalization (DP)**[b] | 52 (25.0%) | 64 (30.8%) | 92 (44.2%) |
| **professional accomplishment (PA)**[c] | 18 (8.7%) | 38 (18.3%) | 152 (73.1%) |
| **Overall burnout**[*] | 4 (1.9%) | 175 (84.1%) | 29 (13.9%) |

[a] Score of $\leq 16$ denoted a low level, a score of 17 to 26 denoted a moderate level, and a score $\geq 27$ denoted a high level.

[b] Score of $\leq 6$ denoted a low level, 7 to 12 denoted a moderate level, and score $\geq 13$ denoted a high level.

[c] Score of $\geq 39$ indicated low level, 32 to 38 denoted a moderate level; $\leq 31$ denoted a high level.

[*] Overall burnout = burnout in all three dimensions (low scores on the EE and DP subscales, with a high score on the PA subscale, denoted a low degree of overall burnout; while high scores on the EE and DP subscales, with a low score on the PA subscale, denoted a high degree of burnout.

levels than resident physicians in the USA, Saudi Arabia (KSA), Taiwan, Nigeria, and Canada (Table 4).

## Discussion

The rate of encountering psychiatric problems such as BOS is increasing day by day among doctors worldwide [42], especially since resident physicians are under intense stress in busy working environments. Within Sudan's collapsed healthcare system and poor infrastructure [43], the burden of providing quality care services with limited medical supplies to a large number of patients can lead to a high rate of BOS [44, 45]. This cross-sectional study was designed to assess the prevalence of BOS and its related factors in a sample of resident physicians in Sudan.

The current study showed that 13.9% of resident physicians who participated in the study have high levels of BOS. This finding suggests a high level of work-related stress among resident physicians in Sudan. According to the findings of a recent multi-country study on the mental health impacts among HCPs in the Eastern Mediterranean Region (EMR), HCPs in Egypt, Iraq, and Sudan demonstrated the highest depression, anxiety, and stress scores compared to those in other countries in EMR [46]. Furthermore, in a recent study that assessed the perceived stress among health professionals in Sudan during the COVID-19 pandemic, more than half of HCPs demonstrated high-stress levels, and most of them had a poor work-family balance [47]. The prevalence rate of BOS in this study was similar to a study conducted among resident physicians in Greece, in which 14.4% of residents met the criteria for BOS [28]. Moreover, a cross-sectional survey in Tunisia concluded that 17.14% of resident physicians had a high level of BOS [33]. Another study in Syria reported a 19.3% prevalence rate of burnout syndrome among resident physicians [40].

When the BOS rate found in this study was compared to that of resident physicians in other countries, most studies detected a higher prevalence of BOS among resident physicians in other countries than the current study results in Sudan. Differences in reported BOS rates in the published literature are related to each country's healthcare system [48]. Another contributing factor may be that most of the studies retrieved and included in the comparison analysis (Table 4) used different tools to measure BOS, and some studies used the short and abbreviated version of the MBI-HSS, which was reported to overestimate BOS prevalence among resident physicians [49]. More importantly, the prevalence rate of BOS found in the present study is similar to other studies of BOS among other clinicians and health personnel in Sudan [50, 51], consistent with our findings.

Significant differences were found in the analysis of parameters related to BOS and its dimensions in this study, with the pediatrics-specialty trainees suffering higher levels

**Table 3. Parameters associated with burnout syndrome and its dimensions among resident physicians in Sudan.**

| Sociodemographic and work-related characteristics | N | Emotional exhaustion | Depersonalization | Personal accomplishment | Overall burnout[a] |
|---|---|---|---|---|---|
| | | Mean ± SD. | Mean ± SD. | Mean ± SD. | Mean ± SD. |
| **Age (years)** | | | | | |
| <30 | 104 | 3.90 ± 1.36 | 2.95 ± 1.31 | 4.06 ± 1.05 | 2.97 ± 0.83 |
| 30 –<35 | 84 | 3.80 ± 1.48 | 2.88 ± 1.28 | 4.28 ± 1.05 | 2.84 ± 0.98 |
| ≥35 | 20 | 3.32 ± 1.51 | 2.47 ± 1.30 | 4.29 ± 1.0 | 2.54 ± 1.10 |
| **F (p)** | | **1.401 (0.249)** | **1.150 (0.319)** | **1.184 (0.308)** | **1.965 (0.143)** |
| **Gender** | | | | | |
| Male | 90 | 3.70 ± 1.35 | 3.03 ± 1.15 | 4.16 ± 1.05 | 2.87 ± 0.90 |
| Female | 118 | 3.88 ± 1.48 | 2.76 ± 1.39 | 4.17 ± 1.04 | 2.88 ± 0.94 |
| **t (p)** | | **0.916 (0.361)** | **1.517 (0.131)** | **0.077 (0.939)** | **0.075(0.940)** |
| **Specialty** | | | | | |
| Medicine | 62 | 3.93 ± 1.23 | 3.05 ± 1.22 | 4.20 ± 1.07 | 2.95 ± 0.87 |
| Pediatric | 38 | 4.36 ± 1.26 | 3.09 ± 1.32 | 4.07 ± 0.97 | 3.19 ± 0.86 |
| Urology | 13 | 3.32 ± 1.34 | 2.66 ± 0.81 | 3.99 ± 0.96 | 2.69 ± 0.71 |
| OBS & Gynae | 26 | 4.27 ± 1.26 | 3.45 ± 1.26 | 4.48 ± 0.91 | 3.08 ± 0.86 |
| Surgery | 38 | 3.84 ± 1.46 | 2.89 ± 1.25 | 4.23 ± 1.20 | 2.87 ± 0.98 |
| ENT | 7 | 3.54 ± 0.90 | 2.46 ± 1.35 | 4.27 ± 0.60 | 2.64 ± 0.78 |
| Oncology | 9 | 3.68 ± 1.07 | 2.40 ± 1.23 | 3.72 ± 0.65 | 2.88 ± 0.80 |
| Dermatology | 13 | 1.46 ± 0.82 | 1.29 ± 0.76 | 4.13 ± 1.0 | 1.57 ± 0.65 |
| Psychiatry | 2 | 2.44 ± 3.14 | 1.0 ± 1.41 | 2.94 ± 3.09 | 2.34 ± 0.48 |
| **F (p)** | | **7.494* (<0.001*)** | **4.727* (<0.001*)** | **0.965 (0.464)** | **4.820*(<0.001*)** |
| **Years of residency experience** | | | | | |
| <3 | 103 | 3.73 ± 1.51 | 2.83 ± 1.35 | 4.09 ± 1.08 | 2.86 ± 0.97 |
| ≥3 | 105 | 3.87 ± 1.34 | 2.92 ± 1.25 | 4.24 ± 1.01 | 2.89 ± 0.88 |
| **t (p)** | | **0.722 (0.471)** | **0.461 (0.645)** | **1.016 (0.311)** | **0.186(0.853)** |
| **Marital status** | | | | | |
| Single | 124 | 3.87 ± 1.35 | 3.0 ± 1.26 | 4.10 ± 1.02 | 2.95 ± 0.83 |
| Married | 81 | 3.75 ± 1.50 | 2.73 ± 1.34 | 4.25 ± 1.09 | 2.79 ± 1.03 |
| Divorced/Widowed | 3 | 2.63 ± 2.29 | 1.73 ± 0.42 | 4.79 ± 0.56 | 1.91 ± 1.22 |
| **F (p)** | | **1.199 (0.304)** | **2.262 (0.107)** | **1.027 (0.360)** | **2.455 (0.088)** |
| **Working hours per duty** | | | | | |
| 8 to11 hours | 37 | 2.72 ± 1.61 | 1.98 ± 1.47 | 3.80 ± 1.14 | 2.36 ± 1.05 |
| 12 to 17 hours | 50 | 3.69 ± 1.13 | 2.77 ± 1.03 | 4.37 ± 0.84 | 2.73 ± 0.74 |
| 18 to 24 hours | 53 | 4.06 ± 1.30 | 3.07 ± 1.16 | 4.13 ± 0.98 | 3.04 ± 0.80 |
| More than 24 hours | 68 | 4.27 ± 1.30 | 3.29 ± 1.25 | 4.26 ± 1.13 | 3.13 ± 0.94 |
| **F (p)** | | **11.979* (<0.001*)** | **9.733* (<0.001*)** | **2.372 (0.072)** | **7.067* (<0.001*)** |

**t:** Student t-test **F:** ANOVA test **N:** Number.

p: p-value for comparing between the different categories.

*: Statistically significant at p ≤ 0.05.

[a] The overall burnout indicates high scores for burnout in all three dimensions.

(Table 3), which could be due to the disparities in the working environments among different medical departments that were suggested to be linked to BOS [9]. Consistent with this finding, many other studies revealed that pediatric-specialty resident physicians with a higher risk of burnout syndrome suffer higher levels [52–56]. Additionally, the burnout rate was positively associated with resident physicians' working hours per duty. This finding is supported by the

**Table 4. Comparison of burnout rate among resident physicians in Sudan and other countries.**

| Author, Year of publication | Country | Sample size | Reported burnout rate | Comparison against the current study |
|---|---|---|---|---|
| Zis, 2014 [28] | Greece | 263 | 14.4% of the residents were found to experience burnout | Slightly higher prevalence rate of burnout |
| Abdulrahman, 2018 [29] | UAE | 302 | 75.5% moderate-to-high EE, 84% had high DP, and 74% had a low sense of PA. | Higher EE, DP, and PA. |
| Afzal, 2010 [30] | USA | 115 | 34% reported high EE, 30% reported high DP, and 24% reported low PA | Lower EE, DP, and PA |
| Agha, 2015 [31] | KSA | 96 | Overall, 88.5% experienced burnout, with high EE in 68.8%, high DP in 63.6%, and low PA in 38.5%. | Higher prevalence rate of burnout, EE, and DP. With lower PA. |
| Al-Dubai, 2013 [32] | Malaysia | 205 | High EE was reported by 36.6% of the respondents. DP and PA were not reported. | Lower EE, |
| Ben Zid, 2018 [33] | Tunisia | 149 | 17.14% had a high level of burnout | Higher prevalence of burnout. |
| Chaukos, 2016 [34] | USA | 68 | 28% of resident physicians had a high level of burnout. | Higher prevalence of burnout. |
| Chen, 2013 [35] | Taiwan | 809 | 13.1% reported high EE. 9.3% reported high DP, and 0.7% reported a high level of decreased PA | Lower EE, DP, and PA. |
| Cheng, 2020 [36] | USA | 105 | 71% prevalence of burnout. | Higher prevalence rate of burnout. |
| Cristina-Nituica, 2021 [37] | USA | 682 | 51% prevalence of burnout. | Higher prevalence rate of burnout. |
| Ofei-Dodoo, 2019 [38] | USA | 218 | 43% prevalence for burnout, 32% reported high EE, 31% high DP, and 22% high level of decreased PA. | Higher prevalence rate of burnout. Lower EE, DP, and PA. |
| Ogundipe, 2013 [39] | Nigeria | 204 | 45.6% had a high level of EE, 57.8% had high DP, and 61.8% with a high level of decreased PA. | Lower EE, DP, and PA. |
| Alhaffar, 2019 [40] | Syria | 3350 | 19.3% prevalence of burnout syndrome. | Higher prevalence of burnout. |
| Ferguson, 2020 [41] | Canada | 718 | 69.4%. prevalence of burnout, with 61.6% high EE, 47.8% high DP, and 29.0% high level of decreased PA. | Higher prevalence of burnout. Lower EE, DP, and PA. |

EE; Emotional exhaustion. DP; Depersonalization. PA; Professional accomplishment.

results of a study conducted among resident physicians in KSA, which reported an association between the resident duty working hours and burnout syndrome [57]. Moreover, there were no significant differences in burnout rates related to resident physicians' sociodemographic factors; age, gender, years of residency experience, and marital status (Table 3). Similar findings have been reported in previous research of BOS among resident physicians in South Africa [58] and the USA [59].

## Strength and limitations

Since this study is a cross-sectional survey, we cannot establish a causal relationship between BOS and the independent variables. The current study has not assessed the level of stress, awareness of coping strategies, or job satisfaction among resident physicians in Sudan to avoid having a lengthy questionnaire. Accordingly, the selection of the explanatory variables was not theoretically supported. In addition, the low response rate of participants may have influenced the findings of this study. Nevertheless, the study provided preliminary evidence regarding the prevalence rate of burnout syndrome among resident physicians in Sudan. Furthermore, the study was limited to Gezira state, due to a lack of sufficient resources for a national survey, limiting the generalization of these study findings to resident physicians in other Sudanese states.

## Conclusion

The study revealed high levels of burnout syndrome among this sample of resident physicians in Sudan, with the pediatrics-specialty resident physicians being especially vulnerable. There

was a significant difference in the level of burnout according to respondents' working hours per duty. Large-scale studies are required to assess the determinants of burnout syndrome among resident physicians in Sudan. In addition, Stakeholders should urgently implement effective remedies to protect the mental health of resident physicians.

## Acknowledgments

We want to thank Prof. Adrian Rabe, Ph.D., for his contributions and assistance in this study.

## Author Contributions

**Conceptualization:** Yasir Ahmed Mohammed Elhadi, Abdelmuniem Ahmed.

**Data curation:** Yasir Ahmed Mohammed Elhadi, Abdelmuniem Ahmed, Elhadi B. Salih, Osman S. Abdelhamed, Mohamed Hayder Hamid Ahmed, Noha Ahmed El Dabbah.

**Formal analysis:** Yasir Ahmed Mohammed Elhadi, Abdelmuniem Ahmed, Elhadi B. Salih, Osman S. Abdelhamed, Mohamed Hayder Hamid Ahmed, Noha Ahmed El Dabbah.

**Investigation:** Yasir Ahmed Mohammed Elhadi, Abdelmuniem Ahmed, Elhadi B. Salih, Osman S. Abdelhamed, Mohamed Hayder Hamid Ahmed, Noha Ahmed El Dabbah.

**Supervision:** Noha Ahmed El Dabbah.

**Writing – original draft:** Yasir Ahmed Mohammed Elhadi, Elhadi B. Salih, Osman S. Abdelhamed, Mohamed Hayder Hamid Ahmed.

**Writing – review & editing:** Yasir Ahmed Mohammed Elhadi, Abdelmuniem Ahmed, Elhadi B. Salih, Osman S. Abdelhamed, Mohamed Hayder Hamid Ahmed, Noha Ahmed El Dabbah.

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
