## [Decision Letter · Decision Letter 0]

14 Feb 2022

PONE-D-21-37637A cross-sectional survey of burnout in a sample of resident physicians in SudanPLOS ONE

Dear Dr. Elhadi,

Thank you for submitting your manuscript to PLOS ONE. After careful consideration, we feel that it has merit but does not fully meet PLOS ONE’s publication criteria as it currently stands. Therefore, we invite you to submit a revised version of the manuscript that addresses the points raised during the review process.

We look forward to receiving your revised manuscript.

Kind regards,

Mehmet Serkan Kirgiz

Academic Editor

PLOS ONE

Journal Requirements:

Additional Editor Comments:

To guide you, I have attached the comments of reviewers. After revising the paper to the comments, the author should resubmit it to the Plos One.

Reviewers' comments:

Reviewer's Responses to Questions

**Comments to the Author**

1. Is the manuscript technically sound, and do the data support the conclusions?

Reviewer #1: Yes

Reviewer #2: Partly

2. Has the statistical analysis been performed appropriately and rigorously? 

Reviewer #1: Yes

Reviewer #2: Yes

3. Have the authors made all data underlying the findings in their manuscript fully available?

Reviewer #1: Yes

Reviewer #2: Yes

4. Is the manuscript presented in an intelligible fashion and written in standard English?

Reviewer #1: Yes

Reviewer #2: No

5. Review Comments to the Author

Reviewer #1: The rate of encountering psychiatric problems such as burnout syndrome and depersonalization is increasing day by day among doctors all over the world, especially since residents are under intense stress in busy working environments.

Reviewer #2: General Comments:

Language needs editing.

Please check for sentences that are repeated more than once.

Specific comments:

Abstract:

- Add detailed data about the participants.

- The results need more details.

- The conclusion does not support the findings. It should be precise and concise. Add the future directions.

Introduction:

- Define "Burnout syndrome" in detail.

- Explain the measured variables.

- You have to focus more on the rationale of the study. What is already known and what will you add to the knowledge.

- Add a clear hypothesis.

Methods:

- How and who administrates the data collection?

- How did you achieve the validity and reliability of the outcome measures?

- Please, reframe the components (SPICES) for methods

i. Study design, setting, sample size

ii. Participant

iii. Intervention/issue of interest (exposure)

iv. Comparison

v. Ethics and endpoint

vi. Statistical analysis

- What were the eligibility criteria for participants?

- How was the sample size determined?

Discussion:

- Introductory paragraph should include the main findings of the study.

- This section needs to be put in the line with objectives and hypotheses.

- Discuss and compare your findings with similar previous studies.

- Explain more the implications and strengths of the study.

- Explain the limitations of the study in detail. The main limitation of the study design is not demonstrated.

Conclusion:

- The conclusion should be precise and concise. Add the future directions.

References:

- Many references are very old. You have to update the references published within the last 5 years.

6. PLOS authors have the option to publish the peer review history of their article (what does this mean?). If published, this will include your full peer review and any attached files.

Reviewer #1: **Yes: **Turgay Dağtekin

Reviewer #2: **Yes: **Walid Kamal Abdelbasset

---

## [Author Response · Author response to Decision Letter 0]

21 Feb 2022

Dear Prof Emily Chenette

Editor-in-Chief of Plos One, 

And 

Mehmet Serkan Kirgiz

Academic Editor Plos One,

Thank you for giving us the opportunity to submit a revised draft of our manuscript titled [A cross-sectional survey of burnout in a sample of resident physicians in Sudan]. We appreciate the time and effort that you and the reviewers have dedicated to providing your valuable feedback on our manuscript. We are grateful to the reviewers for their insightful comments on our paper. We have been able to incorporate changes to reflect most of the suggestions provided by the reviewers. We have highlighted the changes within the manuscript. Here is a point-by-point response to the editors’ reviewers’ comments and concerns.

Editor’s comments and journal requirements 

Response: Dear Editor, the revised manuscript has been formatted to meet PLOS ONE's style requirements. 

2. In your Data Availability statement, you have not specified where the minimal data set underlying the results described in your manuscript can be found.

Response: Data was uploaded to and shared publicly. Data Availability statement was updated to “The datasets used and/or analyzed during the current study are available from the following DOI link: https://doi.org/10.6084/m9.figshare.19184831”

3. Please review your reference list to ensure that it is complete and correct. If you have cited papers that have been retracted, please include the rationale for doing so in the manuscript text, or remove these references and replace them with relevant current references. Any changes to the reference list should be mentioned in the rebuttal letter that accompanies your revised manuscript. If you need to cite a retracted article, indicate the article’s retracted status in the References list and also include a citation and full reference for the retraction notice

Response: The references list was revised according to the editors’ and reviewers comments, references no 1, 3, 19, and 44 in the original submission were removed and updated. All changes were highlighted in the track-changes version. 

Reviewers’ Comments 

First: General Comments

Language needs editing.

Please check for sentences that are repeated more than once. 

Response: Thank you very much for your suggestion, after addressing all reviewer’s concerns, we have sent the manuscript to the language editing and proofreading service, and duplicated parts were removed. 

Second: Specific comments

A- Comments related to the Abstract:

• Add detailed data about the participants.

• The results need more details.

• The conclusion does not support the findings. It should be precise and concise. Add the future directions.

Response: We agree with these remarks, we have revised the Abstract part to include more details about the study participants and future directions for policymakers and areas of further studies. All changes were highlighted. 

B- Comments related to the Introduction:

• Define "Burnout syndrome" in detail.

• Explain the measured variables.

• You have to focus more on the rationale of the study. What is already known and what will you add to the knowledge.

• Add a clear hypothesis. 

Response: Thank you for these comments, we have revised the Introduction part according to the reviewers’ comments. All new parts inserted on page 2 lines 54-59 and page 4 lines 90-95, were highlighted within the revised version of the manuscript. 

C- Comments related to the Methods:

• How and who administrates the data collection?

• How did you achieve the validity and reliability of the outcome measures?

• Please, reframe the components (SPICES) for methods

i. Study design, setting, sample size

ii. Participant

iii. Intervention/issue of interest (exposure)

iv. Comparison

v. Ethics and endpoint

vi. Statistical analysis

• What were the eligibility criteria for participants?

• How was the sample size determined?

Response: Thank you very much for your suggestions to improve the presentation of our work, we agree with most of these remarks. We have provided the required details and revised the Methods part according to the reviewers’ comments. Because the size of the study population (resident physicians at Wad Madani in Gezira state hospitals) was relatively small, we approached and invited members of the target population. We clearly defined the study population instead of sampling in the Methods section. The Cronbach’s Alpha for the MBI-HSS in this study was 0.79 suggesting a high internal consistency of the MBI subscale. The changes were highlighted within the Track-changes version of the Manuscript. 

D- Comments related to the Discussion:

• Introductory paragraph should include the main findings of the study.

• This section needs to be put in the line with objectives and hypotheses.

• Discuss and compare your findings with similar previous studies.

• Explain more the implications and strengths of the study.

• Explain the limitations of the study in detail. The main limitation of the study design is not demonstrated.

Response: We appreciate your kind comments to improve the presentation of our work, we agree with these suggestions. We have provided the required details and revised the Discussion part according to the reviewers’ comments. The changes on page 13 lines 218-223 and page 14 lines 260-267, were highlighted within the track-changes version of the Manuscript. We elaborated more on the strength and limitations of the study. 

E- Comments related to Conclusion :

• The conclusion should be precise and concise. Add the future directions.

Response: We have updated the conclusion part according to the reviewer’s comment. 

F- Comments related to References:

• Many references are very old. You have to update the references published within the last 5 years.

Response: In this study, we performed an online data search to compare burnout rates between resident physicians in Sudan and other countries. We searched the PubMed database for similar cross-specialty studies of burnout among resident physicians published between 2000-2021. Other than these citations, old references were removed and updated.

---

## [Editor Report · Decision Letter 1]

23 Feb 2022

A cross-sectional survey of burnout in a sample of resident physicians in Sudan

PONE-D-21-37637R1

Dear Dr. Elhadi,

We’re pleased to inform you that your manuscript has been judged scientifically suitable for publication and will be formally accepted for publication once it meets all outstanding technical requirements.

Kind regards,

Prof. Dr. Mehmet Serkan Kirgiz

Academic Editor

PLOS ONE
---

## [Editor Report · Acceptance letter]

24 Feb 2022

PONE-D-21-37637R1 

A cross-sectional survey of burnout in a sample of resident physicians in Sudan 

Dear Dr. Elhadi:

I'm pleased to inform you that your manuscript has been deemed suitable for publication in PLOS ONE. Congratulations! Your manuscript is now with our production department. 

Kind regards, 

on behalf of

Professor Dr. Mehmet Serkan Kirgiz 

Academic Editor

PLOS ONE